# Construction of a Form for Users of the Child Welfare System Based on the Delphi Method

**DOI:** 10.3390/children10061026

**Published:** 2023-06-07

**Authors:** Olga Fernández-García, María Dolores Gil-Llario, Rafael Ballester-Arnal

**Affiliations:** 1Department of Developmental and Educational Psychology, Faculty of Psychology, University of Valencia, 46010 Valencia, Spain; olga.fernandez-garcia@uv.es; 2Department of Basic and Clinical Psychology and Psychobiology, Faculty of Health Sciences, Jaume I University, 12007 Castellón de la Plana, Spain; rballest@uji.es

**Keywords:** form, child welfare system, children and adolescents, sociodemographic, sexual abuse, delphi method, instrument, construction

## Abstract

Professionals in charge of designing individualized plans for children and adolescents in the child welfare system often lack the necessary information, either because it has not been systematically collected or because there are doubts about the reliability of the data obtained. The lack of consensual and validated instruments that gather the necessary information has led to the development of a rigorous and effective form, based on the Delphi methodology, aimed at obtaining an exhaustive knowledge of the characteristics of children and adolescents under the child welfare system. Once a consensus of different specialists approved the hetero-informed form, it was completed by 41 professionals working in residential care facilities for 307 children and adolescents. It consists of 66 items grouped into six dimensions: general information, school/work situation, child welfare system history, family visitation history, biological family information, and experiences of sexual abuse. During its construction and validation, a panel of experts analyzed its format and content during the different phases. Most of the items showed good performance, and professionals highlighted their ease of use and relevance. The method used ensured the content validity of this form. This instrument has proven to be a useful and effective tool for collecting sociodemographic information on children and adolescents in the child welfare system, which may improve their conditions.

## 1. Introduction

The inclusion of a minor in the child welfare system is the measure adopted to address a situation of neglect of children or adolescents, intending to provide them with a protective and safe environment that guarantees the complete fulfillment of their rights and the opportunity for complete development [1]. However, according to the latest Statistical Data Bulletin on Child Welfare System, despite existing problems of data homogeneity [2], the number of children and adolescents growing up under the child welfare system in Spain already surpasses 50,000 for the first time [3]. Generally, these minors have experienced several violations of their human rights (right to survival, education, or freedom from any form of violence) and repeated situations of fear and stress that have forced them to develop dissociative mechanisms to survive. This makes it common for them to have established destructive affective models and faces a permanent conflict of belonging during their immersion in the child welfare system [4,5]. For this reason, protection measures must be designed with minors and their needs at the focus, and each minor should be provided with individualized plans that include the specialized resources necessary to meet their differential characteristics [6].

Therefore, it is essential that children and adolescents in foster or residential care be recognized as a group of special vulnerability, with special needs, and that their curricular adaptation and training itineraries are assessed [2]. However, on many occasions, the professional team responsible for developing the individualized intervention plan for the child or adolescent when they first arrive at a residential care facility, as a temporary measure, lacks sufficient information about their situation. Generally, coordination between the different actors and agents involved in the child welfare system is not adequate [2], which means that the professional receives a simple e-mail with general data on the child’s current situation and their evolution in the last facility in which they were attended, or, in the best of cases, they are given an extensive report full of annotations that the social services prepared to assess their situation at a particular time and from which it is difficult to extract concise and useful information. Having to change foster care or residential homes frequently [5,7], as well as a large number of external resources (school resources and external therapists, extracurricular activities, etc.) with which the professional must coordinate [2], makes it difficult to properly transmit information on essential aspects of the child or adolescent.

Additionally, as a result of the in-depth literature review that has been carried out, it is concluded that there is a lack of published instruments designed to collect sociodemographic data that have proven to be valid and effective in the child and adolescent population, and even less so in minors with such specific experiences as those involved in the child welfare system. These forms are often developed ad hoc for a particular research and sample (see [8]). They do not focus on exploring specific aspects of this group (e.g., their history in the child welfare system) that may be of great importance when preparing interventions or explaining other aspects of their psychophysical development.

Given the above, the need for a brief and concise instrument to collect these fundamental aspects while being simple to complete and explore after completion is evident. Furthermore, this would allow data collection on this group to be standardized, achieving more precise statistical reports and making the basic information on minors easily accessible to the different professionals in the child welfare system who work with them. For this purpose, the present study describes the development process and the characteristics of a sociodemographic data form for children and adolescents in the child welfare system, the Child and Adolescent Welfare System Form (CAWSys).

## 2. Materials and Methods

### 2.1. Participants

The participants in this study were 41 professionals working in residential care facilities located in the Eastern of Spain, and who completed the CAWSys of 307 children and adolescents. Specifically, they provided services to children and adolescents who were housed in residential care facilities for general care (70.73%), preparation for emancipation (14.63%), severe behavioral problems (7.32%), or were migrants without known relatives (7.32%). Almost 74% were women, while only 26% were men. Most of them were psychologists (56.1%), while in 21.95% of cases, the form was completed by the director, 12.19% by a social educator, and 9.76% by a social worker. The approach followed was that the professional from the residential care facility who knew the child and their situation best and who had access to the welfare case record to check the necessary information should participate.

In this sense, they reported information on 307 children and adolescents living in residential care facilities. Among 66.1% were male adolescents, while 33.9% were female adolescents, and the mean age was 15.63 (SD = 1.67). Almost half of the sample (48.5%) were between 14 and 16 years old, followed by 39.2% between 17 and 19 years old, and only 12.2% were between 11 and 13 years old. Although the majority of the participants (57.9%) were born in Spain, there were 29.7% who were born in Morrocco, and the remaining nationalities were underrepresented (Eastern European: 4.7%; West African: 3.3%; South/Central American: 2.7%; Pakistani: 1.2%; and Portuguese: 0.6%). Likewise, 27.7% were migrant adolescents without relatives.

### 2.2. Procedure

The Delphi method [9] was used to construct the CAWSys (Figure 1). This method is a well-established approach to answering a research question through the identification of a consensus view across subject experts [9,10]. It is ideal for issues where scientific evidence is absent and it is essential to have information for judgment, as was our case. In this study, gathering the opinions and beliefs of experts from different areas (child welfare system, childhood and adolescence, methodology and research, etc.) was not only useful but also necessary to develop an effective instrument [11]. Also considering the discrepancies and different points of view of experts, while seeking consensus, has allowed us to create a tool that considers the wide variety of existing realities. Moreover, at the methodological level, compared to other techniques, the anonymity of panelists in the survey rounds, the controlled feedback, and the iterative discussions that characterize the Delphi method bring a certain validity to the process of constructing the instrument [12].

*Exploring and defining the theme.* First, an exhaustive literature review of sociodemographic data collection studies and instruments published so far was conducted, and several reports produced by the child welfare system were analyzed, examining their content (areas assessed), format (response format and the number of items), among others. This review was carried out by members of the research team in charge of developing CAWSys, who are experts in the characteristics of the child welfare system, instrument development, and sexual health. As a result of this analysis, six dimensions were considered for inclusion: general information, school/work situation, child welfare system history, family visitation history, biological family information, and experiences of sexual abuse. The first dimension (general information) was included with the aim of obtaining essential information about the child or adolescent, the content of which would allow explaining response patterns in the rest of the dimensions. Secondly, the school/work environment is a setting in which children and adolescents spend a large portion of their time and establish most of their interpersonal bonds, so analyzing their behavior and attitude in this environment can provide essential information. This encouraged the inclusion of items in the school/work situation. Likewise, the inclusion of items on their child welfare system history was strongly justified, as it was one of the main reasons for constructing the form. On the other hand, the experts considered that the importance of the child’s encounters with their relatives (family visitation history) should not be overlooked in their evolution in the child welfare system, as well as the need to explore possible background information of the father and mother that could explain behaviors and attitudes of the child or adolescent (biological family information). Finally, the need to collect information on past victimization experiences of the child or adolescent (experiences of sexual abuse) was considered, given that early knowledge of these events by professionals will determine the individualized intervention protocol.

*Elaboration of the first version of the form, selection of the panel of experts and distribution of the first version of the form to the panel of experts*. Based on this research and the defined dimensions, a preliminary set of items was formulated and shared with an advisory board (or panel) of experts in child and adolescent, in child welfare system (specifically, in the analysis and report writing of children and adolescents in the child welfare system), and in data analysis methodology. The objective was to evaluate the degree of relevance of each item in the construct, thereby increasing the instrument’s content validity by revising the proposed items and suggesting new ones.

*Phase 1*: *Analysis of input and redesign of the second version of the form*. As a result of this analysis, six items were reformulated to improve their wording and comprehension, and six new items were included to collect information on aspects that had not been considered. The resulting document was reviewed again by the group of experts, who ratified its structure.

*Phase 2: Pilot test and analysis of professionals’ notes*. After this final stage, the instrument consisted of 67 items with different response formats (dichotomous, multiple-choice, open-ended, among others), and a psychologist reviewed this provisional version of the CAWSys from a residential care facility who completed the form for five individuals aged between 15 and 18 years, to determine whether the items were correctly understood and if they provided new information. This step allowed further revisions of the items, improving the wording, integrating more inclusive language in one of them, and modifying the response options in five others.

*Preparation of the final form*. Once these improvements had been made, the instrument was definitively established.

In a subsequent phase, the directors of the residential care facilities were contacted to present them with the project and request their collaboration. Next, a member of the group of experts from the research team went to the residential care facilities to train the professional(s) who were going to conduct a CAWSys for each of the children and adolescents in the residential care facility, once the prescriptive consent had been given. Likewise, the pertinent permissions were obtained beforehand from the Directorate General of Childhood and Adolescence within the collaboration agreement signed between the Department of Equality and Inclusive Policies and the SALUSEX research group and the permission granted by the Ethics Committee of the University of Valencia (Spain).

### 2.3. Data Analysis

Descriptive statistics were used for the descriptive analysis of the sample group and the items. All statistical analyses were performed with the IBM SPSS Statistics 23 program.

## 3. Results

### 3.1. CAWSys Description

The instrument consists of 66 items, 59 closed-ended responses (dichotomous and multiple-choice), and 7 open-ended responses, grouped into 6 dimensions that give meaning to the instrument’s structure:General information: This dimension includes nine items that collect basic information about the minor concerning sex assigned at birth, sexual orientation, date of birth, nationality, disability, physical or mental health problems, and psychoactive substance use. Regarding the response format, two items are open-ended (A.3. and A.4.), five are dichotomous (A.1., A.5., A.6., A.7. and A.8.), and two have multiple response options (A.2. and A.9.);School/work situation: Consisting of nine items with multiple response options that collect information about the studies being pursued at the time of the evaluation, if they have started working, academic history, attitude, and school integration;Child welfare system history: The nine items that constitute this dimension collect information on the age of entry into the child welfare system, the reason for this entry, their current legal status, and current and past protection measures. Regarding the response format, three of the items require an open-ended numerical response (C.1., C.4. and C.6.1), two have a dichotomous response format (C.2. and C.5.), while the rest present multiple response options (C.3., C.6., C.6.2 and C.6.3);Family visitation history: This dimension consists of nine items that inquire about the established visitation regime (place, frequency, duration, control, and persons attending), whether it is complied with, and the child’s or adolescent’s assessment of these visits. Four of the items are dichotomous (D.1., D.2., D.3., and D.6.), and the remaining ones have multiple response options (D.4., D.5., D.7., D.8. and D.9.);Biological family information: This dimension aims to inquire about the characteristics of the biological parents of the child or adolescent that may influence or have influenced them and their situation, as well as their relationship with them and their possible siblings. Likewise, some items also include the family’s economic condition, the community environment in which the child or adolescent grew up, and intrafamily relationships. Thus, of the 23 items that make up this dimension, all of them with multiple response options, 9 are duplicated by asking on the one hand about aspects concerning the father and, on the other, about those of the mother;Experiences of sexual abuse: Consisting of seven items, the aim is to inquire about the information available to the residential care facility regarding the possible experiences of sexual abuse experienced by the participant (suspicions, confirmation, frequency, characteristics of the perpetrator, and consequences). Two of the items are open-ended (F.4. and F.6.), while the rest are dichotomous (F.1., F.2., F.3., F.5., and F.7.).

The form should be completed by the professionals of the residential care facilities, based on the reports of the child or adolescent in the child welfare system, and it takes approximately 10 min to complete, depending on the participant’s characteristics.

The final version of the CAWSys is attached in Appendix A.

### 3.2. CAWSys Construction and Item Properties

During the CAWSys construction process, the Delphi method was implemented in two phases. In the first phase, the experts were provided with an initial list of 61 items distributed in six dimensions (see Table 1). After conducting the necessary assessments, they proposed the inclusion of four items exploring the degree of physical and psychological disability of the parents (items E.2.4/E.3.4 and E.2.5/E.3.5) and one item examining the presence of filio-parental violence (item E.1.) in the dimension “Biological family information”. It was also proposed to introduce another item in the dimension “Experiences of sexual abuse” that would consider therapeutic support for those children and adolescents who had suffered sexual abuse (item F.7). Likewise, items A.3. (“Date of birth” instead of “Age”, since it provides more detailed information), A.6. (“Disability” instead of “Disability/functional diversity”, since disability is referred to when the social-service department has granted a degree of disability) and A.9. (“Consumption of psychoactive substances” instead of “Consumption of toxic-dependent substances”, so that sporadic consumption could also be included) of the “General information” dimension were reformulated. On the other hand, the term “minor” was replaced by “minor person” in item C.2. (“Current legal status of the minor person”) of the dimension “Child welfare system history”, in the statement of item D.9. (“Assessment of visits by the minor person in general”) of the dimension “Family visitation history” and in an alternative of item E.4. (“Siblings”) of the dimension “Biological family information”.

In the second phase, the redesigned form was distributed to a residential care facility for completion by its professionals, who made a qualitative assessment of its items by making annotations in the margin to determine argumentative proposals to improve the wording. Firstly, the wording of one of the alternatives in item C.6. (“Current protection measure”) of the dimension “Child welfare system history” was corrected, as it referred to a typology of the residential care facility and did not conform to the new nomenclature. On the other hand, professionals suggested the inclusion of 5 new alternatives in item B.1. (“School/work situation”) of that same dimension, to specify the academic situation of the child and avoid losing information; in item C.6.3 (“Final aim of the intervention”) of the “Child welfare system history” dimension, they also suggested introducing two more precise possible intervention objectives related to the situation of children and adolescents with behavioral problems or who had been in the child welfare system for a short time, in item E.2.9 and E.3.9 (“Employment status”) of the dimension “Biological family information” they detected the need to include an alternative that contemplated the parents’ “Compensation” situation and in item E.1. (“Filio-parental violence”) of the dimension “Biological family information” and F.1. (“Suspected sexual abuse”) of the dimension “Experiences of sexual abuse” the alternative “N/A” was added.

On the other hand, analyzing the percentage of missing responses (unanswered items) (Table 2), it can be stated that, in general, most of the items that constitute CAWSys seem to work adequately, except for some that were not answered by all the participants. This is the case of eight items of the “School/work situation” dimension (items B.2. to B.9.), which address their attitude and integration in school and their academic record and which were left unanswered approximately 9% of the time. To a lesser extent, items C.4. (“Years in the child welfare system”), C.6.1 (“Months enjoying the measure”), and C.6.3 (“Final aim of the intervention”) of the dimension “Child welfare system history” were left blank approximately 5% of the time. Likewise, parent items E.3.7 (“Victim of maltreatment”), E.5. (“Economic situation”), E.6.1 (“Conflicting social dynamics”), E.6.2 (“Presence of a support network”), and E.7. (“Separation/divorce”) were also unanswered by 15, 7, 9, 6 and 20% of the total sample, respectively. In this line, items D.2. to D.9., whose content refers to the established visitation regime, and F.2. to F.7., whose content refers to the characteristics of the alleged sexual abuse experienced, were only to be answered if the item directly preceding them, i.e., items D.1. and F.1., respectively, were answered affirmatively. As a result, the referred items show lower response rates.

### 3.3. Content Validity, Usability, and Relevance of the CAWSys

Content validity was ensured by the consensus reached by the panel of experts. While in the first phase of the process only 62% of the experts reached a consensus, in the second phase 89% of the experts reached a consensus, which is an appropriate percentage to affirm the existence of content validity of the form.

In terms of ease of use, the professionals who participated in the completion of the instrument rated the number and content of the items, as well as the time required to complete them. More than 90% of the experts agreed that 95% of the items could be answered by the tutor or psychologist of the residential care facility, provided that they have access to the child’s or adolescent’s reports. Likewise, 98% of the respondents stated that the items were straightforward to answer due to their format (most of them with multiple response options) and the average time required to complete the form (approximately 10 min).

Regarding the relevance of the CAWSys, 94% of the professionals who participated stated that the CAWSys allowed them to have an overall perspective of the characteristics of the child or adolescent and that it would have been beneficial for them when developing their individualized plan upon arrival at the residential care facility. In 92.2% of the cases, they assured that they would use it to transfer the primary information about the minor. However, 56% suggested that the information on the form was advised to be supported by effective communication between professionals to expand and clarify this information.

## 4. Discussion and Conclusions

CAWSys was developed following a rigorous construction process in which multiple experts from different areas have participated and which has been tested in a real setting (pilot test) before the final format was achieved. This gives sufficient solidity to the construction process, presenting it as a useful and effective instrument for collecting sociodemographic data on children and adolescents in the child welfare system.

Overall, the items performed correctly, since almost 80% were answered correctly by all the participants and the evaluation of the response format was positive. However, the items related to the minor’s academic situation, time in the child welfare system, and those asking about specific aspects of their social and familiar context were not completed by all the professionals. Those who did not complete them alleged that they did not possess such information at the time they filled the forms and that they had to carry out further research (such as asking school personnel, the staff of the residential care facility where the minor had previously resided, and so on) to answer them, which made it difficult for them to respond. In this sense, when using the CAWSys it should be considered that these were the worst performers, as well as excluding the item that asks about parental separation or divorce as it was the item that obtained the worst results. Regarding the response format, the fact that the instrument includes both open-response items, which allow the collection of more detailed information, and closed-response items (dichotomous or multiple-choice), which can be used to obtain more precise data that fits a previously established pattern, thereby reducing the time spent in completing them, appears to be a significant strength of the instrument presented. This allows more subjective data to be collected on relevant and distinctive aspects of each child or adolescent without obtaining highly disparate responses that may limit the subsequent comparison and elaboration of statistics.

In addition, the instrument was found to cover all relevant aspects of the purpose for which it was developed. In other words, according to the panel of experts, all the areas in which it collects information are indispensable and no aspect of particular relevance has been left out. Likewise, the professionals who completed them considered that it was helpful in their task and could be beneficial for managing and transmitting the information.

Regarding the methodology used in the process of constructing this form, the use of the Delphi method, being a flexible technique, helped to encourage, to a greater extent, the reflection and creativity of the experts, a key factor in the elaboration of a much more complete tool. However, above all, it stands out because it made it possible to construct an instrument that considers both the phenomenon and the evaluation context and is mainly oriented towards practical contribution, thanks to the verifiability, comprehensibility, and holism of this method. However, the methodology used also has some limitations in terms of the quality of the evidence reported, as the decisions taken in the process of constructing this instrument have been based mainly on the consensus of the panel of experts and this should be considered. In this regard, it would be advisable for future studies to investigate the validity and efficacy properties of CAWSys, as well as its usefulness in other similar contexts (e.g., judicial system) and in other countries.

Consequently, it can be concluded that the form developed and tested in this study contributes to the transmission of basic information about the child or adolescent of the child welfare system among the professionals working with them. In other words, it contributes to the improvement of interprofessional communication, which would help in the preparation of much more individualized and specialized intervention plans, and on a larger scale, in the compilation of more accurate statistics on the characteristics of this group, avoiding the disparity of the data reported depending on the source consulted. Furthermore, although the main implication of this study is to contribute to the improvement of the work of child welfare professionals, researchers around the world could use this form in their studies to ensure the collection of information on the main socio-demographic aspects of these children, which will help them to make sense of the other variables assessed in their research. This would also help to ensure comparability of data across countries. This is unusual so far and could contribute to improving social policies in countries with the worst statistics, considering the plans implemented in those with better data, with the final aim of optimizing the conditions of these children and adolescents.

## Figures and Tables

**Figure 1 children-10-01026-f001:**
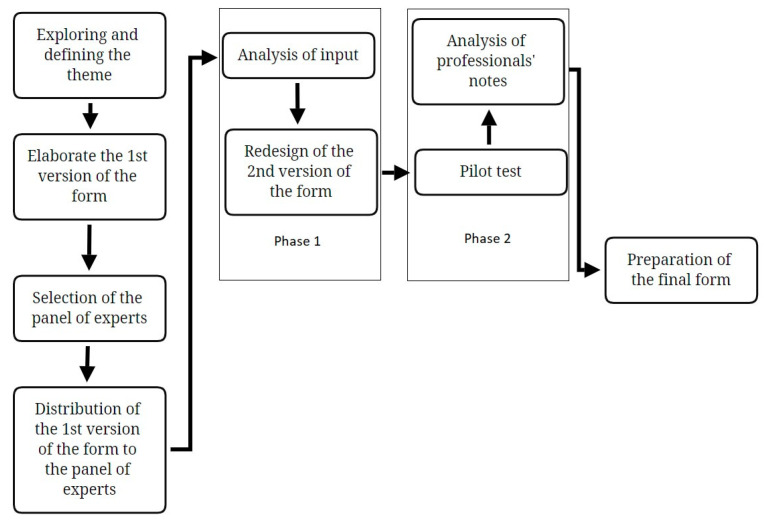
Procedure for the application of the Delphi method.

**Table 1 children-10-01026-t001:** Evolution of the composition of CAWSys in its building process.

Dimensions	Items	First Version	Second Version ^1^	3rd Version ^2^	Final Version
General information	N	9	9	9	9
	Reformulated	NA	3 (items A.3., A.6., A.9.)	0	NA
School/work situation	N	9	9	9	9
	Reformulated	NA	0	1 (item B.1.)	NA
Child welfare system history	N	9	9	9	9
	Reformulated	NA	1 (item C.2.)	2 (items C.6., C.6.3)	NA
Family visitation history	N	9	9	9	9
	Reformulated	NA	1 (item D.9.)	0	NA
Biological family information	N	19	24	24	24
	Added	NA	5 (items E.1., E.2.4, E.2.5, E.3.4, E.3.5)	0	NA
	Reformulated	NA	1 (item E.4.)	3 (items E.1., E.2.9, E.3.9)	NA
	Deleted	NA	NA	NA	1 (item E.7.)
Experiences of sexual abuse	N	6	7	7	7
	Added	NA	1 (item F.7.)	0	NA
	Reformulated	NA	0	1 (item F.1.)	NA
Total of the form	N	61	67	67	66
	Added	NA	6	0	NA
	Reformulated	NA	6	7	NA

Note: NA = Not Applicable, ^1^ After review by the panel of experts, ^2^ After the pilot test.

**Table 2 children-10-01026-t002:** CAWSys form items and the number and percentage of professionals’ responses to each item in relation to the total number of children and adolescents they assessed.

Dimensions	Items CAWSys	N (%)
General information	A.1. Sex assigned at birth:	307 (100%)
	A.2. Sexual orientation:	307 (100%)
	A.3. Date of birth: ^1^	307 (100%)
	A.4. Nationality: ^1^	307 (100%)
	A.5. Unaccompanied migrant child:	307 (100%)
	A.6. Disability:	307 (100%)
	A.7. Physical health problems:	307 (100%)
	A.8. Mental health problems	307 (100%)
	A.9. Consumption of psychoactive substances:	307 (100%)
School/work situation	B.1. Current school/work situation	307 (100%)
	B.2. Has any school adaptation?	280 (91.2%)
	B.3. Integration in the school:	279 (90.8%)
	B.4. Behavior in the classroom:	288 (90.8%)
	B.5. Has been expelled from the school?	280 (91.2%)
	B.6. Attitude and motivation towards learning:	281 (91.6%)
	B.7. School habits/skills:	279 (90.9%)
	B.8. Has repeated a grade?	281 (90.26%)
	B.9. Truancy:	279 (90.8%)
Child welfare system history	C.1. Age of entry into the child welfare system: ^1^	307 (100%)
	C.2. Current legal status of the minor person:	307 (100%)
	C.3. Event giving rise to the placement:	307 (100%)
	C.4. Years in the child welfare system: ^1^	290 (94.46%)
	C.5. Past protection measures:	307 (100%)
	C.6. Current protection measure:	307 (100%)
	C.6.1. Months enjoying the measure: ^1^	292 (95.11%)
	C.6.2. Degree of adaptation/satisfaction with the measure:	307 (100%)
	C.6.3. Final aim of the intervention:	291 (95.1%)
Family visitation history	D.1. Are there established visits?	307 (100%)
	D.2. Are they occurring?	185 (100%)
	D.3. Place of the visits:	185 (100%)
	D.4. Frequency of the visits:	185 (100%)
	D.5. Duration of the visits:	185 (100%)
	D.6. Control of the visits:	185 (100%)
	D.7. Persons with whom the child/adolescent is seen:^1^	185 (100%)
	D.8. Compliance with visits:	185 (100%)
	D.9. Assessment of visits by the child/adolescent (in general):	185 (100%)
Biological family information	E.1. Filio-parental violence:	307 (100%)
	E.2.1/E.3.1 Background in the child welfare system:	M 307 (100%)/F 307 (100%)
	E.2.2/E.3.2 Physical health problems:	M 307 (100%)/F 307 (100%)
	E.2.3/E.3.3 Mental health problems:	M 307 (100%)/F 307 (100%)
	E.2.4/E.3.4 Recognised degree of physical disability:	M 307 (100%)/F 307 (100%)
	E.2.5/E.3.5 Recognised degree of mental disability:	M 307 (100%)/F 307 (100%)
	E.2.6/E.3.6 Substance abuse:	M 307 (100%)/F 307 (100%)
	E.2.7/E.3.7 Victim of maltreatment:	M 307 (100%)/F 264 (85.7%)
	E.2.8/E.3.8 Criminal record:	M 307 (100%)/F 307 (100%)
	E.2.9/E.3.9 Employment status:	M 307 (100%)/F 307 (100%)
	E.4. Siblings:	307 (100%)
	E.5. Economic situation:	286 (93.2%)
	E.6.1 Conflicting social dynamics:	278 (90.5%)
	E.6.2 Presence of a support network:	289 (94.2%)
	E.7. Separation/divorce:	245 (80.1%)
Experiences of sexual abuse	F.1. Suspected sexual abuse:	307 (100%)
	F.2. Confirmation of suspected sexual abuse:	52 (100%)
	F.3. Alleged perpetrator:	52 (100%)
	F.4. Occasions on which it has occurred (approx.): ^1^	52 (100%)
	F.5. Sex of alleged perpetrator:	52 (100%)
	F.6. Short and/or long-term consequences: ^1^	52 (100%)
	F.7. Have you received subsequent therapeutic support?	52 (100%)

Note: M = Mother; F = Father; ^1^ open-ended items.

## Data Availability

The data presented in this study are available on request from the corresponding author upon reasonable request.

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
