# Peer review of "Construction of a Form for Users of the Child Welfare System Based on the Delphi Method"

_children, 2023, doi:10.3390/children10061026_

Round 1
Reviewer 1 Report
First of all, thank you for the exciting paper that can potentially make a valuable contribution to the field. However, I'd suggest adding Confirmatory Factor Analysis (CFA) to your methods that will make your findings and arguments scientifically rigorous.
n/a
Author Response
REVIEWER 1
#0 Comment: First of all, thank you for the exciting paper that can potentially make a valuable contribution to the field. However, I'd suggest adding Confirmatory Factor Analysis (CFA) to your methods that will make your findings and arguments scientifically rigorous.
#0 Answer: We are very grateful that the reviewer considered that our study makes an interesting contribution to the existing literature and to this field of research.
Confirmatory Factor Analysis (CFA) is the most widely used method to assess the accuracy with which different variables measure and evaluate a concept (Morata-Ramírez et al., 2015). However, in our study we do not aim to assess the accuracy with which different variables allow us to obtain information on a construct, but rather to collect descriptive information in a structured way. So, we present a form (a document designed to collect information in a structured way) whose main purpose is to collect socio-demographic data on different aspects of the child or adolescent in a structured way for information and storage purposes.
Also, the application of the CFA, to confirm the structure of an instrument, is not appropriate/convenient in all cases. The CFA requires the fulfilment of certain assumptions that may contradict the nature of the variables and data collected. One of these assumptions was the one postulated by Mulaik (1972) which states that the variables must be continuous. In our case, the variables do not meet this assumption, being discrete (and nominal in some cases), which makes the application of a CFA impossible and inappropriate.
Mulaik, S. A. (1972). The foundations of factor analysis (Vol 88). New York: McGraw-Hill.
Morata-Ramírez, M.ª A., Holgado-Tello, Francisco P., Barbero-García, Isabel, & Mendez, Gonzalo. (2015). Análisis factorial confirmatorio: recomendaciones sobre mínimos cuadrados no ponderados en función del error Tipo I de Ji-Cuadrado y RMSEA. Acción Psicológica, 12(1), 79-90. https://dx.doi.org/doi.org/10.5944/ap.12.1.14362
We thank the reviewer for the opportunity to reflect on this issue.

Reviewer 2 Report
The authors are to be commended for using a sound, social science approach/method to construct an assessment form. Below are some suggestions to strengthen the presentation of the work that the authors have reported in this paper.
Introduction
Lines 62-65 needs a citation to support the authors conclusion (?) that there is a lack of published instruments if other authors have come to the same conclusion. Or state that it is the authors' conclusion based on their review of the the literature.
Materials and Methods
Clarification about what constitutes the study sample is needed. It seems the the "sample" in this study are the 41 professionals working in the residential care facility who completed the form developed through the use of the Delphi method. To describe the sample, any other demographics on these professionals besides their profession should be indicated if available. If not available, then it should be stated that other demographic data was not either collected or available.
It should be stated that the 41 professionals provided services to children and adolescents who were housed in residential care facilities for general care, severe behavioral problems, or were migrants without known relatives. The description of the children and adolescents should be adequate. It does not appear to be necessary to include table 1.
Line 87- please define what is hetero-informed instrument
Lines 93-134, [The description of the measurement Tool- Child and Adolescent Welfare System Form (CAWSys)] should come after description of the Procedures, since engaging in the procedures resulted in the Assessment Tool. The instrument/measurement tool was developed through use of the Delphi Method and is a result/product, not a measurement tool used to conduct the study. This needs to be made clearer in the narrative.
Including the final CAWsys in the Appendix would be helpful to the descriptions provided in the narrative about the tool and its six sections. For example, when. the authors state that in the General Information items, two items are open-ended ,etc., it would be easier to refer to the full instrument and name the item or refer to the item number . The same can be said about School/work situation and other categories.
In describing the Procedure, [line 136 to 189], a brief description of the rationale for using the Delphi Method needs to be included.
Each step of the Delphi Method as graphically presented in Figure 1 would be good to serve as a heading in the narrative to better align the steps in the graphics with the narrative.
Results
Line 255-256, In Table 3, "CAWSys form items, number of participants who responded to each item and percentage of the total sample" --does the N represent 307 participants or responses from 41 professionals using the Measurement Tool to see if it is applicable to the 307 children and adolescents housed at the residential care facility? The Abstract states that it was completed by 41 professionals working in residential care facilities for 307 children and adolescents.
Line 264, "given the disparity in the content of the items" should be explained. Do the authors mean to say that the items are at the categorical level of measurement and thus are not able to be considered for reliability analysis which requires at ordinal level of measurement?
Line 265, the authors need to specify if they are discussing face, content, or some other form of validity. It seems that what is described is relevance of the form as well as ease of using it and fit and useful -- that the items on the form were easily found in the case files or could be found by asking, and they were relevant and perhaps an efficient way to gather data for the assessment of history, present concerns, needs and functioning.
Line 277, is it 56% percent considered or should it be phrased "suggested" or "advocated"
Discussion and Conclusions
Lines 283-285, From a psychometric perspective, more testing would need to occur before reliability and validity would be concluded on the form and the form to be considered reliable and valid.
There needs to be discussion about the strengths and limitations of the methods and analysis that led the construction of the form.
The implications for future research and practice need to be stated.
The English phrasing was not always clear and it would be helpful to double check the phrasing, especially in use of terms such as reliability and validity and the meaning intended by using them.
Author Response
REVIEWER 2
#0 Comment: The authors are to be commended for using a sound, social science approach/method to construct an assessment form. Below are some suggestions to strengthen the presentation of the work that the authors have reported in this paper.
#0 Answer: Thank you for this reviewer's positive comments recognizing the importance of using this merit for the elaboration of a data collection instrument. Of course, we have tried to address all the issues raised by this reviewer and respond to all their concerns, paying special attention to improving the draft manuscript. The reviewer may also find that we have made changes to the English edition of the paper. We thank you in advance for your thorough review of our work and for your concrete suggestions, as we believe that your feedback has been of special relevance to improve our paper.
#1 Comment (Introduction): Lines 62-65 needs a citation to support the authors conclusion(?) that there is a lack of published instruments if other authors have come to the same conclusion. Or state that it is the authors' conclusion based on their review of the the literature.
#1 Answer: As the reviewer will understand, the problem is that there are so few studies on this topic that it is difficult to find published evidence along these lines, although professionals in the field confess this limitation in their daily professional practice in residential care facilities or in the social service administrative agencies, for example. Therefore, this statement is based on our own experience. It was after an arduous and thorough literature review that we realized that there are no standardized forms for the collection of sociodemographic data on children in the child welfare system or, at least, they have not been published. Hence our need and interest in creating this instrument and contributing to the scientific and social community through its publication. We have included the clarifying sentence suggested by the reviewer (line 63-64):
“Additionally, as a result of the in-depth literature review that has been carried out, it is concluded that there is a lack of published instruments…”
We thank the reviewer for their very precise suggestion.
#2 Comment (Materials and Methods): Clarification about what constitutes the study sample is needed. It seems the the "sample" in this study are the 41 professionals working in the residential care facility who completed the form developed through the use of the Delphi method. To describe the sample, any other demographics on these professionals besides their profession should be indicated if available. If not available, then it should be stated that other demographic data was not either collected or available.
#2 Answer: Following the reviewer's recommendation, we have clarified who the participants in this study are and have provided more information on the professionals' place of work and gender. No further demographic data were collected on them.
Line 81-87: “The participants in this study were 41 professionals working in the residential care facilities located in the Eastern of Spain, and who completed the CAWSys about 307 children and adolescents. Specifically, they provided services to children and adolescents who were housed in residential care facilities for general care (70.73 %), preparation for emancipation (14.63 %), severe behavioral problems (7.32 %), or were migrants without known relatives (7.32 %). Almost 74% were women, while only 26% were men.”
We thank the reviewer for their reflection, which has allowed us to improve the “Participants” section.
#3 Comment (Materials and Methods): It should be stated that the 41 professionals provided services to children and adolescents who were housed in residential care facilities for general care, severe behavioral problems, or were migrants without known relatives. The description of the children and adolescents should be adequate. It does not appear to be necessary to include table 1.
#3 Answer: The reviewer's suggested information on professionals has been added (lines 83-86). In addition, the inclusion of the data of the children and adolescents who were assessed has been modified (line 92), and the description has been completed by redacting some of the data in the table (lines 94-100). The table has been deleted on the proposal of the reviewer.
Line 92-100: “In this sense, they reported information on 307 children and adolescents living in residential care facilities. Among 66.1% (n=203) were male adolescents, while 33.9% (n = 104) were female adolescents, and the mean age was 15.63 (SD = 1.67). Almost half of the sample (48.5%) were between 14 and 16 years old, followed by 39.2% between 17 and 19 years old and only 12.2% who were between 11 and 13 years old. Although the majority of the participants (57.9%) were born in Spain, there were 29.7% who were born in Morrocco, and the remaining nationalities were underrepresented (Eastern European: 4.7%; West African: 3.3%; South/Central American: 2.7%; Pakistani: 1.2%; and Portuguese: 0.6%). Likewise, 27.7% were migrant adolescents without relatives.”
#4 Comment (Materials and Methods): Line 87- please define what is hetero-informed instrument.
#4 Answer: When we use the term "hetero-informed instrument" we refer to an instrument to which the informant responds by providing data from another person. In our particular case, the professional will usually respond to the instrument by providing information about the child or adolescent for whom he or she is providing services. In other words, a hetero-informed instrument is a tool that allows us to obtain information about an individual through the information that another person provides about him/her.
However, because the information in the "Participants" section has been corrected, the term "hetero-informed instrument" has been eliminated.
Nevertheless, we thank the reviewer for allowing us to reflect on this term.
#5 Comment (Materials and Methods): Lines 93-134, [The description of the measurement Tool- Child and Adolescent Welfare System Form (CAWSys)] should come after description of the Procedures, since engaging in the procedures resulted in the Assessment Tool. The instrument/measurement tool was developed through use of the Delphi Method and is a result/product, not a measurement tool used to conduct the study. This needs to be made clearer in the narrative.
#5 Answer: We fully agree with the reviewer. In fact, following the reasoning offered by the reviewer, we have decided that the information on the description of CAWSys should be included in the "Results" section, in a new subsection called "CAWSys description" (lines 180-222).
We would like to thank the reviewer for their suggestion as it underlines the quality of our contribution.
#6 Comment (Materials and Methods): Including the final CAWsys in the Appendix would be helpful to the descriptions provided in the narrative about the tool and its six sections. For example, when. the authors state that in the General Information items, two items are open-ended ,etc., it would be easier to refer to the full instrument and name the item or refer to the item number. The same can be said about School/work situation and other categories.
#6 Answer: The reviewer's proposal seems to us to be very appropriate. Therefore, we have created the Appendix I which includes the final version of CAWSys. As the reviewer says, this may help to frame the description of the instrument included in the paper, but we also believe it will be of use and interest to future readers who may be interested in making use of it. In this connection, we have adapted the text of the manuscript, as proposed by the reviewer, citing the item to which we are referring (lines 187-217) and referring the reader to Appendix I (line 222).
#7 Comment (Materials and Methods): In describing the Procedure, [line 136 to 189], a brief description of the rationale for using the Delphi Method needs to be included.
#7 Answer: We fully agree with the reviewer on the need to justify the decision to use this method in our study. Thus, we have included an explanatory paragraph at the beginning of the “Procedure” (lines 103-114):
“This method is a well-established approach to answering a research question through the identification of a consensus view across subject experts [9, 10]. It is ideal for issues that scientific evidence is absent and it is essential to have information for judgement, as was our case. In this study, gathering the opinions and beliefs of experts from different areas (child welfare system, childhood and adolescence, methodology and re-search, etc.) was not only useful but also necessary to develop an effective instrument [11]. Also considering the discrepancies and different points of view of experts, while seeking consensus, has allowed us to create a tool that considers the wide variety of existing realities. Moreover, at the methodological level, compared to other techniques, the anonymity of panelists in the survey rounds, the controlled feedback and the iterative discussions that characterize the Delphi method brings a certain validity to the process of constructing the instrument [12].”
We thank the reviewer for their helpful comment and for the opportunity to provide this justification of a central aspect of our paper.
#8 Comment (Materials and Methods): Each step of the Delphi Method as graphically presented in Figure 1 would be good to serve as a heading in the narrative to better align the steps in the graphics with the narrative.
#8 Answer: The reviewer's proposal seems to us to be a good idea to provide future readers with a more structured text and to improve their understanding of it. For this reason, we have included in the text subtitles in italics that correspond to each of the steps that were followed for the construction of the form (lines 115, 139, 140, 147, 152 and 160), and which are included in the figure 1.
We thank the reviewer for their appropriate suggestion.
#9 Comment (Results): Line 255-256, In Table 3, "CAWSys form items, number of participants who responded to each item and percentage of the total sample" --does the N represent 307 participants or responses from 41 professionals using the Measurement Tool to see if it is applicable to the 307 children and adolescents housed at the residential care facility? The Abstract states that it was completed by 41 professionals working in residential care facilities for 307 children and adolescents.
#9 Answer: The reviewer's reflection is absolutely right. Thanks to this reviewer's comment, we realised that the title of table 3 did not express the idea we wanted to convey. Indeed, table 3 shows the number and percentage of responses that the 41 professionals provided for each item with respect to the total number of children and adolescents they assessed. The title of table 3 reads as follows (lines 281 and 282):
“CAWSys form items, and number and percentage of professionals' responses to each item in relation to the total number of children and adolescents they assessed.”
#10 Comment (Results): Line 264, "given the disparity in the content of the items" should be explained. Do the authors mean to say that the items are at the categorical level of measurement and thus are not able to be considered for reliability analysis which requires at ordinal level of measurement?
#10 Answer: Indeed, the reviewer's reflection is appropriate. Therefore, given the categorical nature of the items and the confusing nature of the information, we decided to remove it. We thank the reviewer for the opportunity to clarify it.
#11 Comment (Results): Line 265, the authors need to specify if they are discussing face, content, or some other form of validity. It seems that what is described is relevance of the form as well as ease of using it and fit and useful -- that the items on the form were easily found in the case files or could be found by asking, and they were relevant and perhaps an efficient way to gather data for the assessment of history, present concerns, needs and functioning.
#11 Answer: One of the strengths of the Delphi Method is that it is a potential method for researching the content validity of the research instruments, though a consensus technique. In this regard, thanks to the review comment, we have included the information about the content validity of the form in this sub-section.
We also believe that with respect to the other information included in this sub-section, the reviewer is correct. Information on the feasibility of using and relevance of CAWSys is provided. Changes have therefore been made to both the title of the sub-section (line 285) and the content of the sub-section (the introductory sentence of each paragraph -lines 290 and 298-) to make it more specific and nuanced.
We thank the reviewer for their accurate suggestion.
#12 Comment (Results): Line 277, is it 56% percent considered or should it be phrased “suggested" or "advocated"
#12 Answer: We thank the reviewer for their suggestion. Indeed, we also consider that it would be advisable to modify these terms for others that would give the sentence a less imposing meaning. In this sense, the following changes have been made (line 303):
“However, 56% suggested that the information on the form was advised to be supported by effective communication between professionals to expand and clarify this information.”
#13 Comment (Discussion and Conclusions): Lines 283-285, From a psychometric perspective, more testing would need to occur before reliability and validity would be concluded on the form and the form to be considered reliable and valid.
#13 Answer: We agree with the reviewer that this statement is too blunt. For this reason, we have rephrased the phrase (lines 310-312):
“This gives sufficient solidity to the construction process, presenting it as a useful and effective instrument for collecting sociodemographic data on children and adolescents in the child welfare system.”
#14 Comment (Discussion and Conclusions): There needs to be discussion about the strengths and limitations of the methods and analysis that led the construction of the form.
#14 Answer: The reviewer is absolutely right in their comment. We agree with them that it is necessary to address in the "Discussion and Conclusions" section the main strengths and weaknesses of the method we have used in the construction of this form, and which is crucial in our study. Therefore, we have added the following information in this section (lines 337-348):
“Regarding the methodology used in the process of constructing this form, the use of the Delphi method, being a flexible technique, has helped to encourage, to a greater extent, the reflection and creativity of the experts, a key factor in the elaboration of a much more complete tool. But, above all, it stands out because it has made it possible to construct an instrument that considers both the phenomenon and the evaluation context and is mainly oriented towards practical contribution, thanks to the verifiability, comprehensibility, and holism of this method. However, the methodology used also has some limitations in terms of the quality of the evidence reported, as the decisions taken in the process of constructing this instrument have been based mainly on the consensus of the panel of experts and this should be considered. In this regard, it would be advisable for future studies to investigate the validity and efficacy properties of CAWSys, as well as its usefulness in other similar contexts (e.g., judicial system) and in other countries.”
#15 Comment (Discussion and Conclusions): The implications for future research and practice need to be stated.
#15 Answer: The last paragraph of the "Discussion and Conclusions" section sets out exactly what implications this study and the development of this instrument have for future practice and research. However, thanks to the reviewer's comment, we realized that we had failed to convey the idea we wanted to convey. We have therefore rewritten some of these ideas and added other implications for future research and practice that we thought it would be interesting to present (lines 349-363).
“Consequently, it can be concluded that the form developed and tested in this study contributes to the transmission of basic information about the child or adolescent of the child welfare system among the professionals working with them. In other words, it contributes to the improvement of interprofessional communication, which would help in the preparation of much more individualized and specialized intervention plans, and on a larger scale, in the compilation of more accurate statistics on the characteristics of this group, avoiding the disparity of the data reported depending on the source consulted. Furthermore, although the main implication of this study is to contribute to the improvement of the work of child welfare professionals, researchers around the world could use this form in their studies to ensure the collection of information on the main socio-demographic aspects of these children and adolescents, which will help them to make sense of the other variables assessed in their research. It would also help to ensure comparability of data across countries. This is unusual so far and could contribute to improving social policies in countries with worst statistics, considering the plans implemented in those with better data, with the final aim of optimizing the conditions of these children and adolescents.”

Round 2
Reviewer 1 Report
Thank you for the explanations on why not to use CFA.
Reviewer 2 Report
Thank you to the authors for their hard work to revising the manuscript and addressing each of the comments and overall feedback provided in the past review. The changes and additions made by the authors are responsive to the issues raised and do provide greater details and explanations which make better understanding of how the study was conducted and results gathered. Appreciate the authors' clearly stating their responses, and where in the manuscript their responses were integrated in the narrative.